# Psychometric Properties of the Turkish Version of the Soft Drink Addiction Scale

**DOI:** 10.3390/nu17010196

**Published:** 2025-01-06

**Authors:** Meryem Kahriman, Murat Baş, Salim Yilmaz

**Affiliations:** 1Department of Nutrition and Dietetics, Faculty of Health Sciences, Acibadem Mehmet Ali Aydinlar University, 34752 Istanbul, Türkiye; murat.bas@acibadem.edu.tr; 2Department of Healthcare Management, Faculty of Health Sciences, Acibadem Mehmet Ali Aydinlar University, 34752 Istanbul, Türkiye; salimyilmaz142@gmail.com

**Keywords:** soft drink, scale, addiction, behavior, validation

## Abstract

Background: Considering the increasing consumption of soft drinks and their adverse health effects, identifying addiction to these drinks in the population is significant. Accordingly, this study aimed to evaluate the reliability and validity of the Turkish version of the Soft Drink Addiction Scale. Methods: For this purpose, we included 669 participants and distributed them homogeneously for exploratory and confirmatory factor analyses. To assess the psychometric properties of the scale, we used the Soft Drink Addiction Scale, the Beverage Intake Questionnaire-15, and a questionnaire that included questions assessing self-efficacy regarding soft drinks and attitudes toward alternatives. Results: The mean age of 669 participants was 34.05 ± 9.26 years. A total of 93.72% were female and 6.28% were male. The scale’s Cronbach’s alpha coefficient was 0.942. An exploratory factor analysis revealed the following three-factor structure: withdrawal syndrome symptoms, persistent desire, and decrease in social and recreational activities, which explained 29.1%, 16%, and 16.9% of the total variance, respectively. A confirmatory factor analysis also confirmed this construct. Moreover, soft drink addiction was associated with self-efficacy and total calories from beverages questioned in the beverage consumption questionnaire. Conclusions: In conclusion, the Turkish version of the Soft Drink Addiction Scale is a reliable and valid tool.

## 1. Introduction

The concept of soft drinks encompasses different beverage categories, including bottled water, carbonated drinks, dilutables, fruit juice, sports and energy drinks, and still and juice drinks [1]. Soft drink consumption is becoming increasingly widespread, and the market continues to reach significant dimensions worldwide. Regarding this issue, the revenue of supermarkets and markets from soft drinks is reported to reach USD 521.20 billion, whereas the revenue from other places, such as restaurants and bars, will reach USD 386.90 billion. Additionally, by 2024, the average per-person volume of the soft drink market is estimated to reach 37.66 L [2]. A report by the Union of Chambers and Commodity Exchanges of Turkiye stated that the aspect wherein households in Turkiye spent the most was food and soft drinks with 24.2% in 2020 [3].

Soft drink consumption has wide-ranging effects and is frequently associated with negative health outcomes [4,5,6,7]. Increased consumption is associated with obesity, type 2 diabetes, ischemic heart disease, and increased mortality [4,6,7]. Furthermore, increased soft drink consumption is associated with adverse oral health outcomes, including increased dental erosion and reduced physical and mechanical properties of the enamel [8]. Generally, the negative effects of soft drinks are attributed to their high sugar content [9]. Addiction is another significant outcome reported regarding soft drinks [10]. Food addiction is not yet recognized as a pathology and is not included in the Diagnostic and Statistical Manual of Mental Disorders-V [11]. However, in contrast, some studies have reported that food addiction is a pathological disorder [12,13]. This finding explains the increase in obesity and suggests that consuming sugar- or fat-rich food can lead to physiological changes similar to using addictive substances, including alcohol, tobacco, and cocaine [14]. Both food and substance addiction have powerful effects mediated by dopamine spikes in the brain’s reward centers. In particularly vulnerable individuals, sudden increases in dopamine can override the brain’s homeostatic control mechanisms, which may be essential in understanding the similarities between substance and food addiction [12]. Attributing addiction to soft drinks, specifically to their high sugar content, is imperative as several of these drinks contain more sugar than the guidelines recommend [9]. Sugar is an essential food that is believed to be addictive owing to its different properties, including overeating, cravings, withdrawal, cross-sensitivity, cross-dependency, reward, and opioid effects [15]. Therefore, to define addiction to food and drinks with high sugar content, studies have been conducted because although the neurobiological basis on which addiction to these foods can develop is known, it has not yet been fully accepted because it does not fully meet the diagnostic criteria for addictive substances, such as overdose [13,16]. Considering the increasing soft drink consumption [2,3] and their negative effects on health, defining this addiction in society and taking necessary precautions are significant [4,5,6,7]. In previous studies, soft drink consumption was assessed using food frequency questionnaires [17,18]. Campos-Ramírez et al. [19] developed the Soft Drink Addiction Scale, based on the Yale Food Addiction Scale, for the first time in the literature to assess soft drink addiction. This scale has not yet been adapted to other languages. In addition, there is no tool with proven reliability and validity for assessing soft drink addiction in Turkey. Therefore, this study aimed to evaluate the reliability and validity of the Soft Drink Addiction Scale, previously developed by Campos-Ramírez et al. [19], in the population of Turkiye. Furthermore, we hypothesized that the Turkish version of this scale can be a reliable and valid tool.

## 2. Materials and Methods

### 2.1. Study Design

This online study was conducted from October to November 2024 using a questionnaire prepared using Google Forms. Participants aged ≥18 years and who volunteered to participate were included in this study. The number of participants was calculated as 210 for the 21-item Soft Drink Addiction Scale, considering the recommendation of 10 times the number of items previously suggested in the literature [20]. However, considering the possibility of data loss, including more participants was planned. Overall, 669 participants were reached. Before study initiation, ethical approval was obtained from Acıbadem Mehmet Ali Aydınlar University Medical Research Ethics Committee (ATADEK-2024/13), and the participants were asked to approve the informed consent form. In addition, this study was conducted in accordance with the principles of the Declaration of Helsinki. To evaluate the reliability and validity of the Turkish version of the Soft Drink Addiction Scale, a questionnaire containing questions about the sociodemographic and general characteristics of the participants, the Soft Drink Addiction Scale [19], the Beverage Intake Questionnaire-15 (BEVQ-15) [21], and a questionnaire inquiring the preference of healthy alternatives to soft drinks and the self-efficacy of soft drinks were used.

### 2.2. Measurements

#### 2.2.1. Soft Drink Addiction Scale

Campos-Ramírez et al. [19] developed the Soft Drink Addiction Scale by converting it from the Yale Food Addiction Scale [22]. The Yale Food Addiction Scale is an essential scale that evaluates food addiction; however, it is not specific to a particular food. Instead, it questions food addiction in general [22,23]. However, addiction can be specific to certain foods. For this reason, Campos-Ramírez et al. [19] developed the Soft Drink Addiction Scale on the basis of the Yale Food Addiction Scale [22,23]. The items in this scale were adapted from “When I start eating certain foods, I end up eating much more than I had planned” to “I find that when I start drinking soft drinks, I end up drinking much more than I had planned”. This scale comprises 21 items and 3 factors as follows: withdrawal syndrome symptoms, persistent desire, and decrease in social and recreational activities. The items are also scored on a 5-point Likert-type scale ranging from “never” to “four or more times per week or daily”. In a previous study involving 394 participants from Mexico, the reliability and validity of this scale were confirmed and it was reported to have good psychometric properties [19].

#### 2.2.2. BEVQ-15

Hedrick et al. [21] developed the BEVQ-15 to assess habitual beverage consumption. This scale includes 15 different categories, including water, 100% fruit juice, milk, soft drinks, tea, coffee, and energy and sports drinks. The participants are asked how frequently they have consumed these beverages in the past month and are scored on a 7-point Likert-type scale ranging from “never” to “at least three times a day.” The participants also report the portions they consume. The total calories consumed from the questioned beverages can be calculated on the basis of the frequency and amount of consumption [21].

#### 2.2.3. Soft Drink Self-Efficacy and Attitude Toward Alternatives

To assess the participants’ self-efficacy toward soft drinks, items were prepared by adapting items from the Eating Self-Efficacy Scale previously developed by Glynn and Ruderman [24]. The scale developed by Glynn and Ruderman [24] contains a total of 25 items, and these items are scored on a 7-point Likert-type scale ranging from “No difficulty controlling eating” to “Most difficulty controlling eating”. For this study, 10 items were adapted, and the items were revised from “I overeat after work or school” to “I drink too many soft drinks after work or school” for soft drinks. Additionally, their preference for healthy alternatives to soft drinks was examined, and the participants were asked to rate these items on a 7-point Likert-type scale ranging from “It is not at all difficult for me to control” to “It is extremely difficult for me to control”. These items included statements, such as “I can stop drinking soft drinks and drink water instead” or “I can stop drinking soft drinks and drink milk/plant-based milk instead”.

### 2.3. Statistical Analysis

Data analyses were performed using the R 4.4.1 program [25]. The participants were homogeneously matched and distributed to the exploratory factor analysis (EFA) and confirmatory factor analysis (CFA) groups according to propensity scores based on sociodemographic and anthropometric measurements. To separate the sample for the exploratory and confirmatory analyses as similar, logistic regression-based propensity scores were calculated for 11 variables, and the process of separating them into two equal groups in space was performed by assigning them to the observations. Similarities were examined using Yates corrected chi-square, likelihood ratio, Pearson chi-square, independent samples *t* test, and Mann–Whitney U tests. The intraclass correlation model in the item analysis was evaluated using the one-way variance analysis model. For EFA, the number of factors was determined by determining the elbow points with a parallel analysis using the principal component and principal axis methods. The Kaiser–Meier–Olkin (KMO) and Bartlett values were used in assumption tests, and the varimax rotation with principal axis factoring method was used in the factor analysis. In the confirmatory section, robust statistics were employed in the modeling to overcome normal distribution and collinearity issues, the robust bifactor-confirmatory factor analysis-restricted maximum-likelihood (CFA MLR) method and Yuan–Bentler correction were interpreted, and the model results were evaluated through the Mplus variant in the calculation. In the reliability analysis, composite reliability (CR), average variance extracted (AVE), discriminant index (27% lower and upper slicing), and Cronbach’s alpha coefficients were used. In the normality testing, skewness and kurtosis values (<|1|) were used. In the hypothesis testing, Mann–Whitney U, Kruskal–Wallis H, and Spearman correlation were used, and Dunn–Bonferroni tests were used as post hoc analysis methods. The evaluations were performed at a 95% confidence level.

## 3. Results

At the end of the study, we reached 669 participants; as we conducted the process using a questionnaire prepared using Google Forms, we did not lose any data and completed the analysis with 669 participants.

Of all the participants, 93.72% (*n* = 627) and 6.28% (*n* = 42) were females and males, respectively. Their mean age was 34.05 ± 9.26 years, and their mean body mass index (BMI) was 24.56 ± 4.76 kg/m^2^. Analyzing the educational status showed that most of the participants (62.78%) had a bachelor’s degree (Table 1).

While EFA (*n* = 339) was conducted with some of the 669 participants, CFA was conducted with the other part (*n* = 330). Subsequently, the propensity scores were calculated to homogeneously distribute the participants according to sociodemographic characteristics and anthropometric measurements; to ensure a homogeneous distribution based on the logistic regression of 23 characteristics (data not shown), 11 numerical and categorical variables (Table 1) were matched and distributed to the EFA and CFA groups according to the propensity scores.

Before proceeding to the EFA, the number of factors was determined using the parallel analysis and revealed three main factors underlying the dataset according to the principal component analysis and principal axis factoring method (Figure 1).

The results of the item analysis revealed the following: the total correlations of the items were at a high level (r > 0.5), the discrimination indices were at medium (0.2–0.39) and high (>0.4) levels, the scale had a high consistency (Cronbach’s alpha = 0.942), and all the items were suitable for EFA without removing any item, considering that the consistency would decrease if any item was deleted (Table 2).

For the EFA, the KMO value obtained from 339 participants was 0.898, and it was accepted that the variables in the dataset had sufficient correlation. Bartlett’s test of sphericity was significant (χ^2^, 6252.001; *df*, 210; *p* < 0.001), and no unit matrix and that the dataset was suitable for factor analysis were concluded. All the items loaded highly (>0.5) on the factors and did not have the issue of loading on more than one factor (ΔLoading > 0.3). The fact that the total variance explained as a result of the analysis was 61.9% showed that the construct validity of the scale in the exploratory part was high and that the items represented the factors well (Table 3).

The factor eigenvalues were 6.103, 3.540, and 3.364 for Factors 1, 2, and 3, respectively. Factors 1, 2, and 3 explained 29.1%, 16.9%, and 16.0% of the total variance, respectively, whereas the total variance explained was 61.9% (Figure 2).

The relationships of four latent variables, including beverage addiction total score, withdrawal syndrome symptoms, persistent desire, and decrease in social and recreational activities, with the different observed variables (B1–B21) are shown in Figure 3. The fact that the factor loadings were relatively high (>1) in some items (B4, B2, B5, B6, B18, and B20) that contained the latent variables showed that using the robust bifactor-CFA was necessary to obtain more reliable results. According to the goodness-of-fit test of the constructed model (14 intrafactor modifications), the Yuan–Bentler corrected chi-square/df value of the model was 1.574 and (256.52/163) the model was a good fit with no significant difference between the observed and model data (*p* = 0.063). The model’s comparative fit index value was 0.947, the Tucker–Lewis index value was 0.932, the root mean square error of approximation value was 0.067, the standardized root mean square residual value was 0.039, the goodness-of-fit index (GFI) value was 0.909, and the adjusted goodness-of-fit index value was 0.857 (Table 4). These values indicate that the model has an acceptable fit.

The standardized factor loadings for the withdrawal syndrome symptom, persistent desire, and decrease in social and recreational activity factors ranged from 0.583 to 0.903, 0.586 to 0.833, and 0.620 to 0.854, respectively, and were highly significant (*p* < 0.001). In the general model, standardized factor loadings had a range of 0.337–0.872, and all the items had significant loadings (*p* < 0.05). For CR, all the factors scored at a high level (≥0.7). The AVE value was slightly low (0.481) for the persistent desire factor and high (≥0.5) for the others. For all the factors, Cronbach’s alpha reliability coefficient was highly reliable (≥0.8) (Table 4).

The participants’ scores on the withdrawal syndrome symptoms, persistent desire, decrease in social and recreational activities subscales, and the total scale were 12.53 ± 5.9, 9.65 ± 5.01, 5.42 ± 1.78, and 27.59 ± 10.41, respectively (data not shown).

No statistically significant difference was observed in the Soft Drink Addiction Scale in terms of the participants’ sex, educational level, and physical activity for at least 150 min per week (*p* > 0.05). In contrast, a significant difference was noted in the persistent desire score in terms of occupation (χ^2^, 21.198; *p* = 0.002 **) and marital status (z, −3.409; *p* = 0.001 **). Furthermore, smokers showed higher withdrawal syndrome symptom scores (z, −2.495; *p* = 0.013 *) (Table 5).

Examining the relationships between the scales showed a strong positive relationship between the total score of the Soft Drink Addiction Scale and self-efficacy-1 (r = 0.624; *p* < 0.001 ***) and a strong negative relationship with self-efficacy-2 (r = −0.537; *p* < 0.001 ***). Moreover, a moderate positive correlation was observed between the Soft Drink Addiction Scale total score and total beverage kcal from the beverage consumption questionnaire (r = 0.364; *p* < 0.001 ***) (Figure 4).

## 4. Discussion

Food with high sugar and fat contents can lead to addiction [15]. Soft drinks are also an essential food group whose addiction is debated, especially owing to their sugar content [9]. However, a tool with confirmed reliability and validity for the population of Turkiye is not yet available. Therefore, we aimed to evaluate the reliability and validity of the Turkish version of the Soft Drink Addiction Scale. First, we evaluated the reliability of the scale and the number of factors using a parallel analysis. The reliability of a scale can be assessed by testing internal consistency by determining whether the scale items contribute to the structure being measured [26]. Different methods, including Cronbach’s alpha or corrected item–total correlation, can be used for assessing internal consistency [27]. In our study, the corrected item–total correlation was >0.5, and the scale Cronbach’s alpha value was 0.942. In the literature, the corrected item–total correlation has been recommended to be >0.30 [28]. Regarding Cronbach’s alpha, ≥0.9, 0.9–0.8, and 0.8–0.7 indicate excellent, good, and acceptable reliability, respectively [29]. In the original scale developed by Campos-Ramírez et al. [19], the Cronbach’s alpha value of the scale was 0.903. In this direction, our scale provides internal consistency and reliability conditions similar to the original scale. Regarding the number of factors, parallel analysis has been recommended as one of the best methods [30]. Consequently, in our study, we determined the number of factors using the parallel analysis method and noted that the scale comprised three factors. Subsequently, we evaluated the KMO value for EFA and observed it to be 0.898. To have good acceptability, the KMO value must be >0.70 [31]. This value indicates that our sample is sufficient for factor analysis.

Evaluating the factor loadings for the EFA showed that all the items had high loadings (>0.5) on the factors they formed. The literature has stated that the minimum acceptable value for factor loading is 0.30 [32]. Accordingly, our scale meets this qualification. Furthermore, EFA revealed the following three-factor structure explaining 61.9% of the total variance: withdrawal syndrome symptoms, persistent desire, and decrease in social and recreational activities. The withdrawal syndrome symptom, persistent desire, and decrease in social and recreational activity factors explained 29.1%, 16%, and 16.9% of the total variance, respectively. Moreover, a CFA confirmed this three-factor structure. Similarly, these three structures were confirmed in the original scale, and this structure explained 54.8% of the total variance [19]. These findings indicate that the Turkish version of the Soft Drink Addiction Scale is a reliable and valid tool for assessing withdrawal syndrome symptoms, persistent desire, and decrease in social and recreational activities.

Evaluating the scores of the participants revealed that the total score was 27.59 ± 10.41. Considering that the highest score that can be obtained from the scale is 105.00, it may be concluded that our population does not have a high level of addiction. Previous studies have suggested that individuals’ soft drink addiction can differ according to sociodemographic variables [10,33,34]. Soft drink consumption has been reported to be more prevalent in males, younger individuals, those with lower educational levels, and those who smoke and drink alcohol [10,33,34]. In our study, no significant differences were noted for soft drink addiction in terms of sex, educational level, and physical activity status. However, significant differences were observed for the total or subfactor scores in terms of occupation, marital status, smoking status, and lifestyle. These findings highlight the need to identify groups at risk of soft drink addiction according to sociodemographic factors and develop specific recommendations. Additionally, notably, this scale has been validated in the general population, and its validity and reliability should be considered for specific groups.

Self-efficacy is a concept that plays a significant role in addictive behaviors. It influences the initial development of addictive habits and the process of behavior change, including the cessation of such habits and maintenance of abstinence [35]. In this study, soft drink addiction was strongly positively correlated with the self-efficacy scale score, which we constructed using items such as “I drink a lot of soft drinks after school or work.” Furthermore, soft drink addiction was strongly negatively correlated with the scores of the scale that we constructed using items such as “I can stop drinking soft drinks and drink water instead” wherein we assessed self-efficacy toward alternative beverages. Similarly, Campos-Ramírez et al. [19] reported significant relationships between the subfactors of the Soft Drink Addiction Scale and self-efficacy. These findings confirmed that self-efficacy can be effective in the initiation and maintenance of addictive behaviors as previously stated in the literature [35]. Additionally, these significant correlations emphasize the construct validity of the Soft Drink Addiction Scale. Moreover, a positive correlation was observed between soft drink addiction and total beverage kcal from the beverage consumption questionnaire. This finding confirms that the consumption of these drinks will increase as addiction increases.

This study had some limitations. First, it was conducted online, and all the data were based on the participants’ self-reports. Second, our population comprised mainly females and individuals with a high educational level. Thirdly, the BMI means of the participants were within the normal range. Therefore, the findings of this study represent people with normal BMI. However, overweight or obese individuals are likely to consume more soft drinks and become addicted. Fourth, our study was conducted with adult participants only. Considering the developing soft drink market, it is expected that these drinks affect pediatric populations, especially adolescents. Therefore, the evaluation of the reliability and validity of this scale for specific populations should be emphasized.

## 5. Conclusions

To the best of our knowledge, this is the first study to evaluate the reliability and validity of the Soft Drink Addiction Scale developed by Campos-Ramírez et al. [19] in the population of Turkiye. Accordingly, we confirmed that this scale measures the following three factors: withdrawal syndrome symptoms, persistent desire, and decrease in social and recreational activities. Furthermore, we determined that soft drink addiction was associated with self-efficacy and calorie intake from beverage consumption. Therefore, the Soft Drink Addiction Scale is a reliable and valid instrument for the population of Turkiye. In addition, it can be a practical tool for identifying soft drink addiction in the general population and developing measures to address it.

## Figures and Tables

**Figure 1 nutrients-17-00196-f001:**
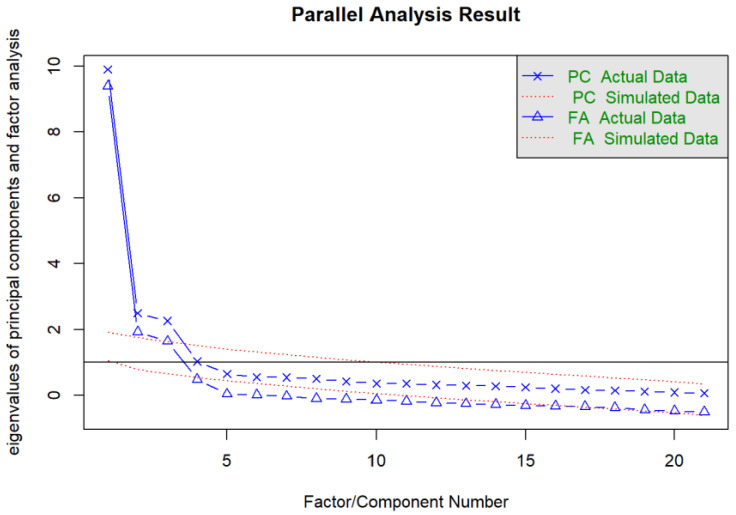
Determining the number of factors with parallel analysis.

**Figure 2 nutrients-17-00196-f002:**
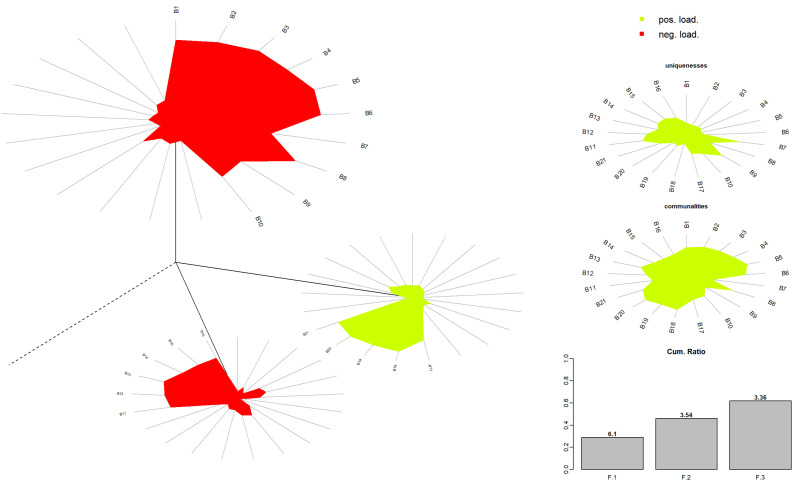
Exploratory factor analysis dandelion plot.

**Figure 3 nutrients-17-00196-f003:**
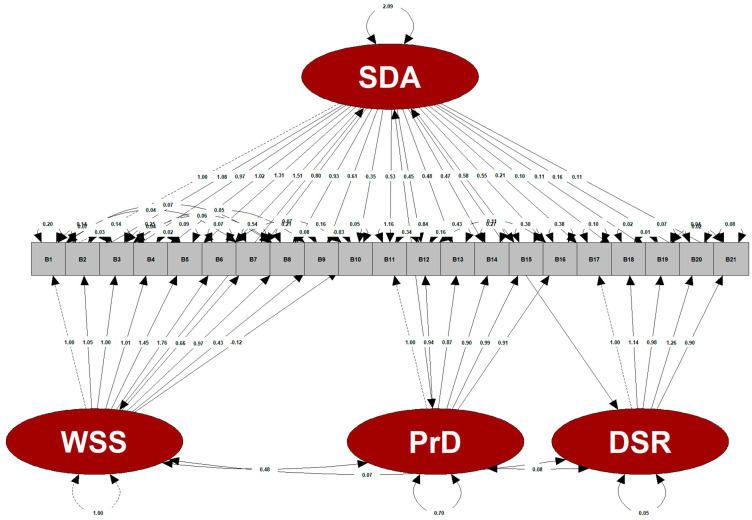
Confirmatory Analysis Path Diagram.

**Figure 4 nutrients-17-00196-f004:**
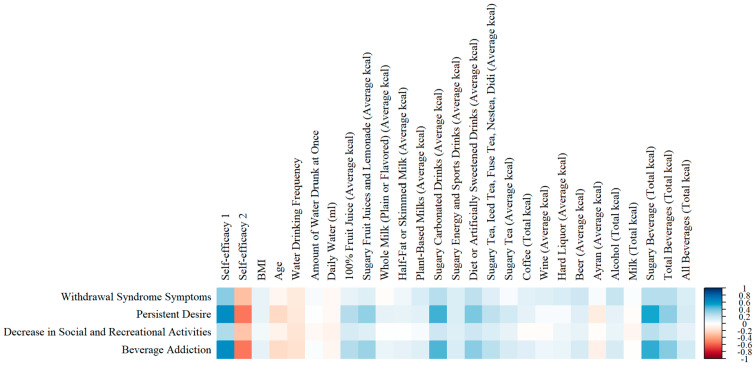
The relationship between Soft Drink Addiction Scale, self-efficacy scale for soft drinks, BMI, age, water consumption, and caloric beverage consumption.

**Table 1 nutrients-17-00196-t001:** Participants’ characteristics.

	IA and EFA Group (*n* = 339)		CFA Group (*n* = 330)		*p*	Total	
	*n*	%	*n*	%		*n*	%
Sex							
Female	314	92.63	313	94.85	0.305 _a_	627	93.72
Male	25	7.37	17	5.15		42	6.28
Educational Level							
Primary school	2	0.59	0	0.00	0.464 _b_	2	0.30
Secondary school	2	0.59	2	0.61		4	0.60
High school	23	6.78	18	5.45		41	6.13
Bachelor’s degree	214	63.13	206	62.42		420	62.78
MSc/PhD	98	28.91	104	31.52		202	30.19
Occupation							
Civil servant	68	20.06	68	20.61	0.214 _c_	136	20.33
Retired	16	4.72	7	2.12		23	3.44
Housewife	22	6.49	11	3.33		33	4.93
Self-employed	35	10.32	45	13.64		80	11.96
Private sector employee	116	34.22	117	35.45		233	34.83
Student	54	15.93	53	16.06		107	15.99
Unemployed	28	8.26	29	8.79		57	8.52
Marital Status							
Single	150	42.25	155	46.97	0.529 _c_	305	45.59
Married	189	58.75	175	53.03		364	54.41
Smoking Status							
Yes	96	28.32	84	25.45	0.454 _c_	180	26.91
No	243	71.68	246	74.55		489	73.09
150 min exercise per week							
Yes	138	40.71	124	37.58	0.453 _c_	262	39.16
No	201	59.29	206	62.42		407	60.84
Lifestyle							
I am single and live with my child/children	9	2.65	11	3.33	0.430 _c_	20	2.99
I live with roommates	10	2.95	13	3.94		23	3.44
I am not married and live with family members	95	28.02	98	29.70		193	28.85
I live with a spouse/partner, and we have a child	145	42.77	121	36.67		266	39.76
I live with a spouse/partner, and we have no children	45	13.27	58	17.58		103	15.40
I live alone	35	10.32	29	8.79		64	9.57
	Min–Max	$\bar{x}$ ± s Q_2_ (Q_1_–Q_3_)	Min–Max	$$\bar{x}$$ ± s Q_2_ (Q_1_–Q_3_)			
Age (years)	18–65	34.74 ± 9.77 34 (27–40)	19–63	33.35 ± 8.66 32 (27–39)	0.051 _d_	18–65	34.05 ± 9.26 33 (27–40)
Body weight (kg)	38–130	68.38 ± 15.24 65.00 (56.50–76.00)	43–116	66.47 ± 13.49 63.00 (57.00–74.00)	0.168 _e_	38–130	67.44 ± 14.43 65 (57–75)
BMI (kg/m^2^)	13.46–42.97	24.90 ± 4.93 23.99 (21.45–27.68)	17.10–42.87	24.22 ± 4.55 23.36 (20.77–27.03)	0.064 _e_	146–195	165.55 ± 6.88 165 (160–170)
						13.46–42.97	24.56 ± 4.76 23.73 (21.11–27.25)

_a_, Yates correction; _b_, likelihood ratio; _c_, Pearson chi-square; _d_, independent samples *t* test; _e_, Mann–Whitney U test; IA, item analysis; EFA, exploratory factor analysis; CFA, confirmatory factor analysis.

**Table 2 nutrients-17-00196-t002:** Item analysis and reliability scores.

Item No.	CITC	Μ ± Σ	DI (%27)	Alpha If Deleted Any Item
1	0.674	1.27 ± 0.79	0.935	0.926
2	0.711	1.22 ± 0.70	0.793	0.925
3	0.712	1.21 ± 0.67	0.728	0.925
4	0.770	1.30 ± 0.82	1.043	0.924
5	0.773	1.31 ± 0.83	1.109	0.924
6	0.709	1.23 ± 0.70	0.837	0.925
7	0.463	1.41 ± 0.84	0.837	0.930
8	0.674	1.22 ± 0.65	0.652	0.926
9	0.622	1.19 ± 0.63	0.598	0.927
10	0.734	1.22 ± 0.69	0.761	0.925
11	0.519	2.11 ± 1.28	2.272	0.932
12	0.483	1.61 ± 1.14	1.728	0.931
13	0.645	1.45 ± 0.95	1.391	0.926
14	0.593	1.43 ± 0.98	1.326	0.928
15	0.651	1.53 ± 1.00	1.533	0.926
16	0.685	1.52 ± 1.01	1.554	0.926
17	0.595	1.12 ± 0.54	0.435	0.928
18	0.602	1.07 ± 0.39	0.272	0.929
19	0.558	1.10 ± 0.47	0.337	0.929
20	0.587	1.10 ± 0.52	0.370	0.928
21	0.522	1.08 ± 0.43	0.283	0.929

CITC, corrected item–total correlation; DI, discriminant index; Cronbach’s alpha, 0.942; F = 50.586; *p* < 0.001; *n* = 349.

**Table 3 nutrients-17-00196-t003:** Exploratory factor analysis results.

	English and Turkish Versions of the Items	Factor 1 (Withdrawal Syndrome Symptoms)	Factor 2 (Decrease in Social and Recreational Activities)	Factor 3 (Persistent Desire)
I1	My behavior toward my soft drink consumption causes me stress or distress./Alkolsüz içecek tüketimime yönelik davranışlarım strese veya sıkıntıya neden oluyor.	−0.806	0.185	−0.115
I2	I have experienced anxiety when I decrease or avoid drinking soft drinks./Alkolsüz içecekleri azalttığımda veya içmekten kaçındığımda kaygı yaşadım.	−0.819	0.199	−0.186
I3	I experience agitation or any other physical symptoms when I decrease or avoid drinking soft drinks./Alkolsüz içecekleri azalttığımda veya içmekten kaçındığımda ajitasyon veya diğer fiziksel semptomlar yaşıyorum.	−0.844	0.161	-
I4	I drink the same types and amounts of soft drinks even when they cause me emotional trouble./Bana duygusal sıkıntı yarattıklarında bile aynı türde ve miktarda alkolsüz içecek içiyorum.	−0.819	0.135	−0.26
I5	I drink soft drinks to the point where I feel bad physically./Fiziksel olarak kötü hissettiğim noktaya kadar alkolsüz içecekler içiyorum.	−0.853	0.106	−0.283
I6	I must increase the regular amount of soft drinks because I have to feel satisfied./Memnun hissetmem için alkolsüz içecek miktarını artırmam gerekiyor.	−0.834	-	−0.212
I7	I feel exhausted after drinking soft drinks in excess./Fazla alkolsüz içecekler içtikten sonra kendimi yorgun hissediyorum.	−0.560	-	-
I8	My soft drink consumption has caused me depression, anxiety, anger, or guilt./Alkolsüz içecek tüketimim depresyona, kaygıya, öfkeye veya suçluluğa neden oldu.	−0.792	0.182	-
I9	My soft drink consumption has caused a health problem or made one worse./Alkolsüz içecek tüketimim bir sağlık sorununa neden oldu veya daha da kötüleştirdi.	−0.546	0.150	−0.161
I10	I have drunk soft drinks to avoid sensations of agitation or any other physical symptoms I live with./Ajitasyon hissinden veya birlikte yaşadığım diğer cziksel semptomlardan kaçınmak için alkolsüz içecekler içtim.	−0.613	0.208	−0.308
I11	I drink soft drinks with most of the meals I have during the day./Gün içinde yediğim öğünlerin çoğuyla birlikte alkolsüz içecekler içerim.	−0.113	-	−0.646
I12	My soft drink consumption is high; however, I do not consider it an issue./Alkolsüz içecek tüketimim yüksek ama bunu bir sorun olarak görmüyorum.	−0.155	-	−0.687
I13	I drink soft drinks even when I am not thirsty anymore./Artık susamadığım zamanlarda bile alkolsüz içecekler içerim.	−0.136	0.200	−0.747
I14	When soft drinks are not available, I attempt to obtain them even if I have other options, such as regular or flavored water./Alkolsüz içecekler mevcut olmadığında, normal veya aromalı su gibi başka seçeneklerim olsa bile onları almaya çalışıyorum.	−0.128	0.272	−0.664
I15	When I am drinking soft drinks, I end up having more than I had planned./Alkolsüz içecekler içerken, planladığımdan daha fazlasını alıyorum.	−0.191	0.211	−0.659
I16	I experience a significant craving or urgency to consume soft drinks when I have decreased or avoided its consumption./Alkolsüz içecek tüketimini azalttığımda veya tüketimden kaçındığımda büyük bir özlemim veya isteğim var.	−0.211	0.208	−0.685
I17	I avoid certain social/professional situations because there will not be soft drinks available./Bazı sosyal/profesyonel desteklerden kaçınıyorum çünkü alkolsüz içecekler olmayacak.	−0.197	0.665	−0.278
I18	My soft drink consumption is such that I stop performing activities, including working, spending time with my family/friends, and other activities I like./Alkolsüz içecek tüketimim nedeniyle çalışmak, ailemle/arkadaşlarımla vakit geçirmek ve sevdiğim aktiviteleri yapmayı bırakıyorum.	−0.227	0.848	−0.189
I19	I have avoided certain family, social, or professional situations where there will be soft drinks available because I am afraid of drinking in excess./Fazla içmekten korktuğum için alkolsüz içeceklerin bulunacağı belirli aile, sosyal veya profesyonel desteklerden kaçındım.	−0.19	0.815	−0.185
I20	My soft drink consumption is such that I experience depression, anxiety, anger, or guilt in such a way that I stop performing activities, including working, spending time with my family/friends, or other activities I like./Alkolsüz içecek tüketimim nedeniyle depresyon, kaygı, öfke veya suçluluk duyuyorum, bu nedenle çalışmak, ailemle/arkadaşlarımla vakit geçirmek veya sevdiğim diğer aktiviteler gibi aktiviteler yapmayı bırakıyorum.	−0.272	0.819	−0.134
I21	I experience issues with my work and school skills, family, or social activities owing to my soft drink consumption./Alkolsüz içecek tüketimim nedeniyle iş ve okul becerilerim, ailem veya sosyal aktivitelerimle ilgili sorunlar yaşıyorum.	-	0.767	−0.188

I, item; KMO, 0.898; χ^2^, 6252.001; *df*, 210; *p* < 0.001; principal axis factoring; explained total variance, 0.619.

**Table 4 nutrients-17-00196-t004:** Maximum likelihood robust CFA model fit indices.

Fit Index and Thresholds Used			Analysis Value		
χ^2^/*df* ≤ 5.00			2.446		
χ^2^/*df* using the Yuan–Bentler correction ≤ 2.00			1.574		
0.90 ≤ CFI ≤ 1.00			0.936		
0.90 ≤ Robust CFI ≤ 1.00			0.947		
0.90 ≤ TLI ≤ 1.00			0.918		
0.90 ≤ Robust TLI ≤ 1.00			0.932		
RMSEA < 0.08			0.042		
Robust RMSEA < 0.08			0.067		
sRMR < 0.08			0.039		
0.85 ≤ GFI ≤ 1.00			0.909		
0.85 ≤ AGFI ≤ 1.00			0.857		
Item	Standardized Factor Loading Bifactor S Model				Standardized Factor Loading Bifactor G Model
	Withdrawal Syndrome Symptoms	Persistent Desire		Decrease in Social and Recreational Activities	Soft Drink Addiction
I1	0.780 ***	-		-	0.767 ***
I2	0.852 ***	-		-	0.838 ***
I3	0.814 ***	-		-	0.811 ***
I4	0.755 ***	-		-	0.767 ***
I5	0.900 ***	-		-	0.872 ***
I6	0.903 ***	-		-	0.863 ***
I7	0.583 ***	-		-	0.583 ***
I8	0.730 ***	-		-	0.727 ***
I9	0.711 ***	-		-	0.720 ***
I10	0.766 ***	-		-	0.755 ***
I11	-	0.586 ***		-	0.383 ***
I12	-	0.594 ***		-	0.372 ***
I13	-	0.726 ***		-	0.465 ***
I14	-	0.794 ***		-	0.504 ***
I15	-	0.833 ***		-	0.563 ***
I16	-	0.781 ***		-	0.574 ***
I17	-	-		0.674 ***	0.536 ***
I18	-	-		0.819 ***	0.381 **
I19	-	-		0.651 ***	0.338 *
I20	-	-		0.854 ***	0.451 **
I21	-	-		0.620 ***	0.337 **
Factors		CR		AVE	CA
Withdrawal Syndrome Symptoms		0.967		0.750	0.943
Persistent Desire		0.845		0.481	0.872
Decrease in Social and Recreational Activities		0.976		0.894	0.848
Beverage Addiction		0.954		0.518	0.917

AGFI, adjusted goodness-of-fit index; CFI, comparative fit index; TLI, Tucker–Lewis index; RMSEA, root mean square error of approximation; I, item; SFL, standardized factor loading; S, sub; G, general; CR, composite reliability; AVE, average variance of explained; CA, Cronbach’s alpha; SRMR, standardized root mean square residual; * *p* < 0.05; ** *p* < 0.01; ***: *p* < 0.001.

**Table 5 nutrients-17-00196-t005:** Comparison of participants by sociodemographic characteristics according to their Soft Drink Addiction Scale scores.

	*n*	*WSS*	*PD*	*DSRA*	SDA
		Q_2_ (Q_1_–Q_3_)	Q_2_ (Q_1_–Q_3_)	Q_2_ (Q_1_–Q_3_)	Q_2_ (Q_1_–Q_3_)
Sex					
Female	627	10.00 (10.00–12.00)	8.00 (6.00–11.00)	5.00 (5.00–5.00)	23.00 (21.00–29.00)
Male	42	10.00 (10.00–10.00)	9.50 (6.00–15.50)	5.00 (5.00–5.00)	24.50 (21.00–33.00)
*z*		−1.512	−1.815	−0.837	−0.793
*p*		0.130	0.070	0.403	0.428
Educational Level					
High school and below	47	10.00 (10.00–13.00)	7.00 (6.00–13.00)	5.00 (5.00–5.00)	23.00 (21.00–31.50)
Bachelor’s degree	420	10.00 (10.00–12.00)	8.00 (6.00–11.00)	5.00 (5.00–5.00)	23.00 (21.00–30.00)
MSc and PhD	202	10.00 (10.00–12.00)	7.50 (6.00–11.00)	5.00 (5.00–5.00)	23.00 (21.25–29.00)
*χ* ^2^		1.485	0.115	1.756	0.077
*p*		0.476	0.944	0.416	0.962
Post hoc		-	-	-	-
Occupation					
Student	107	10.00 (10.00–12.00)	9.00 (7.00–14.00)	5.00 (5.00–5.00)	26.00 (22.50–34.00)
Private sector employee	233	10.00 (10.00–12.00)	8.00 (6.00–11.00)	5.00 (5.00–5.00)	23.00 (22.00–29.00)
Unemployed	57	10.00 (10.00–13.00)	8.00 (6.00–12.00)	5.00 (5.00–5.00)	24.00 (21.00–35.00)
Self-employed	80	10.00 (10.00–11.25)	7.00 (6.00–11.00)	5.00 (5.00–5.00)	23.00 (21.00–28.25)
Civil servant	136	10.00 (10.00–11.00)	7.00 (6.00–10.00)	5.00 (5.00–5.00)	23.00 (21.00–26.25)
Housewife	33	11.00 (10.00–12.00)	7.00 (6.00–11.00)	5.00 (5.00–5.00)	23.00 (21.00–30.00)
Retired	23	10.00 (10.00–10.00)	6.00 (6.00–8.50)	5.00 (5.00–5.00)	21.00 (21.00–23.50)
*χ* ^2^		10.586	21.198	9.409	23.277
*p*		0.102	0.002 **	0.152	<0.001 ***
Post hoc		-	a > b, c, d, e, f, g b > g	-	a, b, c, d, e, f > g a > b, d, e
Marital Status					
Single	305	10.00 (10.00–12.00)	8.00 (6.00–13.00)	5.00 (5.00–5.00)	24.00 (21.00–32.00)
Married	364	10.00 (10.00–12.00)	7.00 (6.00–10.00)	5.00 (5.00–5.00)	23.00 (21.00–27.00)
*z*		−0.640	−3.409	−1.541	−2.995
*p*		0.522	<0.001 ***	0.123	0.003 **
Smoking Status					
Yes	180	10.00 (10.00–13.00)	8.00 (6.00–11.00)	5.00 (5.00–5.00)	24.00 (21.75–31.25)
No	489	10.00 (10.00–11.00)	8.00 (6.00–11.00)	5.00 (5.00–5.00)	23.00 (21.00–29.00)
*z*		−2.495	−0.506	−0.541	−1.450
*p*		0.013 *	0.613	0.589	0.147
150 min exercise per week					
Yes	407	10.00 (10.00–12.00)	8.00 (6.00–11.00)	5.00 (5.00–5.00)	23.00 (21.00–29.00)
No	262	10.00 (10.00–12.00)	7.00 (6.00–11.00)	5.00 (5.00–5.00)	23.00 (21.00–30.00)
*z*		−0.183	−0.675	−0.203	−0.352
*p*		0.855	0.500	0.839	0.725
Lifestyle					
I am not married and live with family members._a_	193	10.00 (10.00–11.00)	8.00 (6.00–13.00)	5.00 (5.00–5.00)	24.00 (22.00–32.00)
I live with my roommates _b_	23	10.00 (10.00–11.50)	9.00 (6.00–15.50)	5.00 (5.00–5.00)	25.00 (21.00–36.50)
I live with a spouse/partner, and we have no children _c_	103	10.00 (10.00–13.00)	8.00 (6.00–11.50)	5.00 (5.00–5.00)	24.00 (22.00–30.50)
I live alone _d_	64	10.00 (10.00–12.25)	8.00 (6.00–12.25)	5.00 (5.00–5.00)	24.50 (21.00–32.00)
I live with a spouse/partner, and we have a child _e_	266	10.00 (10.00–11.75)	7.00 (6.00–9.75)	5.00 (5.00–5.00)	23.00 (21.00–27.00)
I am single and live with my child/children _f_	20	10.00 (10.00–11.00)	6.50 (6.00–8.00)	5.00 (5.00–5.00)	22.00 (21.00–23.00)
*χ* ^2^		2.041	22.984	4.731	20.025
*p*		0.844	<0.001 ***	0.450	0.001 **
Post hoc		-	a, b, c, d > e, f	-	a, c, d, e > f a, c > e

*z*, Mann—Whitney U test; χ^2^, Kruskal—Wallis H test; Post hoc, Dunn—Bonferroni test; *: *p* < 0.05; **: *p* < 0.01; ***: *p* < 0.001.

## Data Availability

The data presented in this study are available on request from the corresponding author due to ethical reasons.

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
