# Peer review of "Psychometric Properties of the Turkish Version of the Soft Drink Addiction Scale"

_nutrients, 2025, doi:10.3390/nu17010196_

Round 1
Reviewer 1 Report
Comments and Suggestions for Authors
The publication aims to assess the usefulness of a questionnaire examining the scale of addiction to soft drinks. The growing consumption of non-alcoholic beverages and the specific trend of consuming non-alcoholic beverages suggests the need for this type of research. The authors assessed the reliability of the Turkish version of the Soft Drink Addiction Scale. 669 participants took part in the study. To determine psychometric properties, the Soft Drink Dependence Scale and the Beverage Consumption Questionnaire-15 were used.
The Cronbach's alpha coefficient for the scale was 0.942. Exploratory factor analysis revealed: withdrawal symptoms, persistent cravings, and a decline in social and recreational activity. The Turkish version of the Soft Drink Addiction Scale could be a reliable and valid tool.
The article was prepared correctly. In table 3, in the first column, the description is in English and Turkish. Why? Do you need a table for the description or is it enough to do it in text form? Similarly with Table 5, most of the parameters are described in the text.
The examined tool may be useful, but research would be required on appropriately selected and diverse groups. Then the assessment of the tool would be more reliable. The study included random people, although a description of the participants was made, but the evaluation of the tool requires more numerous and defined groups of participants.
Author Response
Point 1: The publication aims to assess the usefulness of a questionnaire examining the scale of addiction to soft drinks. The growing consumption of non-alcoholic beverages and the specific trend of consuming non-alcoholic beverages suggests the need for this type of research. The authors assessed the reliability of the Turkish version of the Soft Drink Addiction Scale. 669 participants took part in the study. To determine psychometric properties, the Soft Drink Dependence Scale and the Beverage Consumption Questionnaire-15 were used.
The Cronbach's alpha coefficient for the scale was 0.942. Exploratory factor analysis revealed: withdrawal symptoms, persistent cravings, and a decline in social and recreational activity. The Turkish version of the Soft Drink Addiction Scale could be a reliable and valid tool.
Response 1: Dear reviewer, thank you for your contribution. In line with your suggestion, we revised our manuscript and uploaded it to system.
Point 2: The article was prepared correctly. In table 3, in the first column, the description is in English and Turkish. Why? Do you need a table for the description or is it enough to do it in text form?
Response 2: In the first column of Table 3, the Turkish and English versions of the scale items are given. We thought of presenting them this way so that people who read the article can see the scale items clearly. We also gave them in a table because it would be more complicated if we gave them in the text.
Point 3: Similarly with Table 5, most of the parameters are described in the text.
Response 3: In Table 5, we did not show all statistical values in the text. A brief explanation of Table 5 is included in the text. The reason we present this data in detail as Table 5 is to ensure that readers can see all the data in detail.
Point 4: The examined tool may be useful, but research would be required on appropriately selected and diverse groups. Then the assessment of the tool would be more reliable. The study included random people, although a description of the participants was made, but the evaluation of the tool requires more numerous and defined groups of participants.
Response 4: We mentioned the situation you mentioned at the end of the Discussion section as 'Limitations'. Our population consisted mainly of women and highly educated people. Therefore, we think it would be a good idea to conduct validity and reliability studies in different populations.
Reviewer 2 Report
Comments and Suggestions for Authors
Thank you for the opportunity to review this study entitled “Psychometric Properties of the Turkish Version of Soft Drink Addiction Scale” (nutrients-3380568).
The paper focused on exploring the reliability and validity of the Turkish version of the Soft Drink Addiction Scale. Participants were 669 respondents.
In my opinion, the research topic is relevant, and the study is interesting. The large size of the sample is a significant strength of the paper. In parallel, some issues need to be addressed before the paper will be suitable for publication.
· Abstract: Please, add information about the sample (Mean age, SD, Percentage of men and women) to provide a clear picture of what will be presented in the paper.
· Introduction: a broader overview of existing tools on this topic at the international level should be provided. Following this, it should be explained why it was decided to focus on the Soft Drink Addiction Scale.
· Introduction: the authors stated: “Furthermore, we hypothesized that the Turkish version of this scale can be a reliable and valid tool”. This could be a good hypothesis, but it is not clear what justifies it. Why do you make these assumptions? Had the original study already demonstrated good psychometric properties? It should be made explicit.
· • Method: “[…] and these participants were homogeneously matched and distributed to 76 the exploratory factor analysis (EFA) and confirmatory factor analysis (CFA) groups agreeing to propensity scores based on sociodemographic and anthropometric measures.” This text should be moved within the “data analysis” section.
· • Method: the recruitment processes, as well as the inclusion and exclusion criteria, are unclear. Were non-soft drinkers also included? Or was a question included to exclude these subjects from completing the questionnaire?
· Measurements: please add information about the internal consistency in the present sample of all the used measures.
· The limitations section should be enriched and associated with suggestions for future research.
· The “Conclusions” section should be enriched by further discussing the practical implications of the results of this research.
Best wishes
Author Response
Point 1: The paper focused on exploring the reliability and validity of the Turkish version of the Soft Drink Addiction Scale. Participants were 669 respondents.
In my opinion, the research topic is relevant, and the study is interesting. The large size of the sample is a significant strength of the paper. In parallel, some issues need to be addressed before the paper will be suitable for publication.
Response 1: Dear reviewer, thank you for your contributions. In line with your suggestion, we revised our manuscript and uploaded it to system.
Point 2: Abstract: Please, add information about the sample (Mean age, SD, Percentage of men and women) to provide a clear picture of what will be presented in the paper.
Response 2: Taking your suggestion into account, we added information about participant characteristic to the Abstract section (lines 17, 18).
Point 3: Introduction: a broader overview of existing tools on this topic at the international level should be provided. Following this, it should be explained why it was decided to focus on the Soft Drink Addiction Scale.
Response 3: Considering your suggestion, we explained that in previous studies, soft drink consumption was assessed with food consumption frequency questionnaires and that the tool measuring addiction was first developed by Campos-Ramirez et al. We aimed to test the validity of this scale since there is no scale developed for this purpose in Turkey (lines 64-69).
Point 4: Introduction: the authors stated: “Furthermore, we hypothesized that the Turkish version of this scale can be a reliable and valid tool”. This could be a good hypothesis, but it is not clear what justifies it. Why do you make these assumptions? Had the original study already demonstrated good psychometric properties? It should be made explicit.
Response 4: We have already explained in the 'Discussion' section that the original scale has good psychometric properties. Taking your suggestion into account, we have also stated this in the Method section (lines 115-117).
Point 5: Method: “[…] and these participants were homogeneously matched and distributed to 76 the exploratory factor analysis (EFA) and confirmatory factor analysis (CFA) groups agreeing to propensity scores based on sociodemographic and anthropometric measures.” This text should be moved within the “data analysis” section.
Response 5: Taking your suggestion into account, we have moved this sentence to the Data Analysis section.
Point 6: Method: the recruitment processes, as well as the inclusion and exclusion criteria, are unclear. Were non-soft drinkers also included? Or was a question included to exclude these subjects from completing the questionnaire?
Response 6: We planned to conduct the study with adult participants. Accordingly, we excluded participants under the age of 18. We stated this in the Methods-Study Design section. We did not have any other exclusion criteria. Considering the dietary habits of Turkish society, it is almost impossible not to be exposed to soft drinks. Therefore, we did not set not consuming soft drinks as an exclusion criterion.
Point 7: Measurements: please add information about the internal consistency in the present sample of all the used measures.
Response 7: Cronbach's alpha values reflecting the internal consistency of the scale are shown in Table 2.
Point 8: The limitations section should be enriched and associated with suggestions for future research.
Response 8: In line with your suggestion, we enriched Limitations section.
Point 9: The “Conclusions” section should be enriched by further discussing the practical implications of the results of this research.
Response 9: We took your suggestion and wrote about its practical benefits.